# IL-1 Family Members in Bone Sarcomas

**DOI:** 10.3390/cells13030233

**Published:** 2024-01-25

**Authors:** Lorena Landuzzi, Francesca Ruzzi, Evelin Pellegrini, Pier-Luigi Lollini, Katia Scotlandi, Maria Cristina Manara

**Affiliations:** 1Experimental Oncology Laboratory, IRCCS Istituto Ortopedico Rizzoli, 40136 Bologna, Italy; evelin.pellegrini@ior.it (E.P.); katia.scotlandi@ior.it (K.S.); mariacristina.manara@ior.it (M.C.M.); 2Laboratory of Immunology and Biology of Metastasis, Department of Medical and Surgical Sciences (DIMEC), University of Bologna, 40126 Bologna, Italy; francesca.ruzzi2@unibo.it

**Keywords:** tumor microenvironment, immune escape, macrophages, cytokines, osteosarcoma, Ewing sarcoma, bone metastasis, IL-1 family, IL-1RAP, chimeric-antigen-receptor-modified effectors

## Abstract

IL-1 family members have multiple pleiotropic functions affecting various tissues and cells, including the regulation of the immune response, hematopoietic homeostasis, bone remodeling, neuronal physiology, and synaptic plasticity. Many of these activities are involved in various pathological processes and immunological disorders, including tumor initiation and progression. Indeed, IL-1 family members have been described to contribute to shaping the tumor microenvironment (TME), determining immune evasion and drug resistance, and to sustain tumor aggressiveness and metastasis. This review addresses the role of IL-1 family members in bone sarcomas, particularly the highly metastatic osteosarcoma (OS) and Ewing sarcoma (EWS), and discusses the IL-1-family-related mechanisms that play a role in bone metastasis development. We also consider the therapeutic implications of targeting IL-1 family members, which have been proposed as (i) relevant targets for anti-tumor and anti-metastatic drugs; (ii) immune checkpoints for immune suppression; and (iii) potential antigens for immunotherapy.

## 1. Introduction

Interleukin (IL)-1 was originally recognized as an endogenous pyrogen. Subsequently, many related ligand and receptor molecules, constituting the IL-1 family, have been identified, revealing the fine regulation and intricate crosstalk between the immune response, inflammation, and several physiological processes [1].

IL-1 family members are also highly implicated in tumors, playing a role in tumorigenesis, angiogenesis, invasion, and metastatic progression. They also have a prominent role in the induction of the IL-17-mediated response and in shaping the tumor microenvironment (TME), driving the recruitment of myeloid-derived suppressor cells (MDSCs) and tumor-associated macrophages (TAMs), leading to tumor immune escape [2].

In this review, we will focus on the multiple roles of IL-1 family members in bone sarcomas, mainly osteosarcoma (OS) and Ewing sarcoma (EWS), which are highly metastatic tumors, and in bone metastasis. We will also explore new therapeutic options targeting IL-1 pathways and using IL-1 family members as targets for receptor-modified immune effectors. OS and EWS are the most common bone sarcomas in children and adolescents and are characterized by a high risk of metastasis. Thanks to combined therapeutic approaches, patients with localized disease at diagnosis have achieved a 65–70% 5-year survival rate, but in the case of metastatic disease, the survival rate barely reaches 30% [3,4]. The identification of the mechanisms responsible for the metastatic dissemination of sarcoma cells remains a major challenge to improve the clinical management of these tumors.

## 2. The IL-1 Family Landscape

The IL-1 family includes eleven soluble ligands with agonist, antagonist, or anti-inflammatory properties and ten main receptor chains. Namely, the soluble molecules with agonistic activity are IL-1α, IL-1β, IL-18, IL-33, IL-36α, IL-36β, and IL-36γ; the receptor antagonists are IL-1Ra, IL-36Ra, and IL-38; the cytokine with an anti-inflammatory function is IL-37 (for a comprehensive and updated review of IL-1 family members and a single cell analysis of their activities, see Mantovani et al. 2019 [1] and Cui et al. 2023 [5]).

An accurate database of immune response to cytokines was built by administering 86 different cytokines to mice, collecting their lymph nodes, and performing single cell RNA sequencing (scRNA-Seq) in 17 immune cell types. The study revealed that the complexity of cytokine responses and the plasticity of immune cells are much greater than previously thought. IL-1 family members are active in nearly all lymphoid and myeloid subpopulations, induce highly cell-type-specific transcriptomic responses, and act as drivers of polarization in most immune cell types [5]. Members of the IL-1 family, cytokines and receptors, and their major activities are summarized in Table 1.

### 2.1. IL-1 Family Receptor Complexes

Seven signaling receptor complexes, deriving from the combination of different chains, and other soluble receptor chains can be recognized, forming a complex landscape that ensures the strict regulation of this system. Major details are reported as follows.

The IL-1 receptor 1 (IL-1R1, also named IL-1RI or CD121a) coupled with IL-1R3, also known as the IL-1R accessory protein (IL-1RAcP or IL-1RAP), can bind IL-1α and IL-1β, inducing a positive signal, while negative regulation is triggered by the binding of the antagonists IL-1Ra and IL-38. The soluble IL-1R1 (sIL-1R1) also provides negative regulation.

IL-1R2, also called IL-1RII or CD121b (associated with IL-1RAP on the cell membrane), and soluble IL-1R2 (sIL-1R2) are decoy receptors that bind IL-1β and provide negative regulation.

IL-1R4 (also named ST2, T1, DER4, Fit-1, IL-33Rα) coupled to IL-1RAP is the receptor for IL-33 and provides positive signaling upon binding. IL-1R5 (also known as IL-18Rα) coupled to IL-1R7 (also known as IL18Rβ) is the activating receptor for IL-18, highly expressed on NK cells. IL-18BP is a high-affinity decoy receptor for both IL-18 and IL-37.

IL-1R6 (also named IL-1Rrp2 or IL-36R) coupled with IL-1RAP is the receptor for IL-36α, β, γ, triggering positive regulation, and for IL-36Ra and IL-38, providing negative regulation.

IL-1R8 (also named single immunoglobulin IL-1R-related receptor, SIGIRR, or TIR8) coupled to IL-1R5 (IL-18Rα) is the receptor for IL-37. For the IL-1R8 single chain, IL-1R9 (also known as IL-1 receptor accessory protein-like 1, IL-1RAPL1, or named TIGIRR-2) and IL-1R10 (also known as IL-1 receptor accessory protein-like 2, IL-1RAPL2, or named TIGIRR-1) corresponding ligands are unknown [1]. Of note, IL-1RAPL1 (or IL-1R9) was previously named IL-1R8 and IL-1R10 was previously known as IL-1R9 [11]. In some databases, such as Online Mendelian Inheritance in Man (OMIM) and Gene by the National Institute of Health (NIH), IL-1RAPL1 is also reported as IL-1R8; therefore, some discrepancies in nomenclature may occur [11]. IL-1R9 and IL-1R10 are supposed to have a role in brain development and function [11].

Upon ligand binding, the IL-1 receptor chains will dimerize and then recruit the adaptor protein named myeloid differentiation primary response gene 88 (MyD88) through the intracellular Toll-like/IL-1R (TIR) domain, forming the myddosome complex. This protein activates downstream kinases such as IL-1R-associated kinases (IRKAs) and tumor necrosis factor receptor-associated factor 6 (TRAF6). The final mediators of their activity include activator protein-1 (AP-1), c-Jun N-terminal kinase (JNK), p38, mitogen-associated protein kinases (MAPKs), extracellular signal-regulated kinases (ERKs), members of the interferon-regulatory factor (IRF), and nuclear factor-kB (NF-kB) [12].

However, the picture becomes more complicated with the presence of different isoforms. IL-1RAP has four alternatively spliced variants, including the full-length transmembrane chain, the soluble isoform (sIL-1RAP) produced after the cleavage of the extracellular domain and mainly released by the liver. sIL-1RAP binds to IL-1α or -β, increasing their affinity for IL-1R2 and favoring negative regulation. sIL-1RAP-β is a shorter soluble chain, while IL-1RAPb is an isoform found only in neuronal cells, but it is unable to recruit canonical downstream signaling proteins. IL-1RAPb can modulate the expression of neuronal genes upon exposure to IL-1, in addition to acting as trans-synaptic cell adhesion molecules in the brain to form synapses, and it also plays a role in neuronal cell survival [12,13]. Finally, several genetic polymorphisms of the IL-1RAP gene have been identified, some of which play a role in several chronic inflammatory diseases [14].

### 2.2. IL-1 Family Cytokines

IL-1 family cytokines are differentially expressed in tissues and we simply summarize their main features here. IL-1α is constitutively expressed in epithelial, endothelial, and stromal or mesenchymal cells but can also be produced by all myeloid cells [1]. The IL-1α precursor is active without enzymatic processing but shows increased activity after cleavage. IL-1α on macrophages is mainly present as an integral membrane protein and acts in a juxtracrine manner, activating IL-1R1 and recruiting its co-receptor IL-1R3 on adjacent cells. Moreover, in the nucleus, the IL-1α precursor can directly bind DNA and act as a transcription factor [1]. IL-1α acts locally and is not detected in blood. In contrast, IL-1β is not constitutively expressed, is produced mainly by myeloid cells, and can be detected systemically in the circulation in response to damage and inflammation. IL-1β production requires two signals, the first inducing the transcription of the IL1B gene, followed by the activation signal, leading to the activation of inflammasome complexes and inflammatory caspases, such as caspase-1, to cleave the IL-1β precursor into mature IL-1β [15]. The relevance of inflammasomes in cancer has been excellently analyzed in two recent reviews [16,17]. The inflammasome is a cytosolic protein complex assembling nucleotide-binding domain and leucine-rich repeat protein 3 (NLRP3), apoptosis-associated speck-like protein containing a caspase recruitment domain (ASC), and procaspase-1. NLRP3 inflammasome activation leads to the processing and secretion of the pro-inflammatory cytokines IL-1β and IL-18 [16], and polymorphisms or mutations in NLRP3 have been linked to different types of cancer [17].

Emergency hematopoiesis requires IL-1α and IL-1β, which are induced locally in the bone marrow niche in response to damage and play a role in the differentiation and activation of myelomonocytic cells for innate immunity. IL-1β is the major driver of bone marrow niche remodeling and hematopoietic aging [18]. IL-1 is involved in prolonging the survival of neutrophils and monocytes and is required in the regulation of the Th17 or Type 3 immune response by acting on Th17 and γδ T cells.

IL-1β plays a key role in the TME. By scRNA-Seq, Caronni and collaborators [19] discovered, in pancreatic ductal adenocarcinoma, the peculiar signature of the IL-1β-positive TAM subpopulation (IL1B+ TAM signature), a subset of macrophages enriched in genes of the inflammatory response (IL1B, TNF, NLRP3, PTGS2), of leukocyte recruitment (CXCL1, CXCL2, CCL3), and of angiogenesis induction (VGFA, THBS1, PDGFB), but depleted in interferon response and antigen presentation genes. The expression of the IL1B+ TAM signature in RNA-seq data from The Cancer Genome Atlas (TCGA) was associated with poor patient survival. Prostaglandin E2 (PGE2) and TNF were the TME factors able to elicit the IL-1β+ TAM population with an inflammatory loop between tumor cells and IL-1β-expressing TAM, sustaining tumor growth.

IL-1β, by inflammasome activation, is critical also to produce IL-22 by T helper 22 (Th22) cells, a cancer-promoting cytokine directly induced by some cancer cells [20].

The receptor antagonist IL-1Ra (also named IL-1RN) is produced by monocytes, macrophages, neutrophils, fibroblasts, and epithelial cells. The soluble form sIL-1Ra is produced by hepatocytes. As a competitive inhibitor of IL-1R1, IL-1Ra reduces the effects of both IL-1α and IL-1β, inhibits the physiological responses elicited by both cytokines, and stops their secretion, as well as the production of IL-2 and its receptor. A recombinant form of IL-1Ra is commercially available and is known as anakinra, commonly used to treat arthritis and recently repurposed also for anti-cancer therapies against bone metastases to antagonize the many pro-tumor activities of IL-1β [21].

IL-33 plays a key role in type 2 innate and adaptative immunity by modulating the activities of Th2, Treg, and M2 macrophages. IL-33 is mainly secreted by hematopoietic, stromal, and parenchymal cells through unconventional mechanisms or upon cell necrosis. IL-33 plays a protective role against parasites by promoting the recruitment, amplification, and maturation of neutrophils and eosinophils. It can exist in three forms: full-length IL-33 (FLIL-33), secreted soluble cytokine IL-33 (sIL-33), and nuclear IL-33 (nIL-33), with different implications in tumor immunity that can vary depending on the tumor histotype [22,23,24].

IL-18 is a driver of type 1 innate and adaptative immunity. It is produced by monocytes, macrophages, osteoblasts, and keratinocytes and stimulates IFN-γ production by Th1 and NK cells. IL-18 mediates cytotoxic T cell activation, while promoting IgG2 production in B cells. Defective IL-18-mediated regulation of NK cells may be responsible for increased metastatic growth [1,25]. Its action is at least partially counteracted by IL-37, which competes for the same receptor and mainly mediates NK cell suppression. IL-37 is an anti-inflammatory cytokine that translocates to the nucleus but also transmits a signal via a surface membrane receptor [7,8].

IL-36 cytokines include three agonistic cytokines, IL-36α, IL-36β, and IL-36γ, previously termed IL-1F6, IL-1F8, and IL-1F9, and an antagonist, IL-36Ra, previously named IL-1F5. IL-36 agonistic cytokines are produced by epithelial cells and act on dendritic cells (DCs) to enhance their maturation and cytokine production, playing a key role at the interface between the innate and adaptive immune responses [9,26]. IL-38 binds to the same receptor as IL-36 but exerts antagonist effects and downregulates the Th17 response [10].

All the IL-1 family members have been implicated in both pro-tumor- and anti-tumor-related mechanisms; two exhaustive reviews are given by Sun et al., 2022 and Litmanovich et al., 2018 [6,27].

## 3. IL-1 Family in Osteosarcoma

Under physiological conditions, members of the IL-1 family can affect bone formation and bone remodeling in different ways. IL-1α can induce osteoblast apoptosis and inhibit osteoblastogenesis by activating the JNK/p38 MAPK pathway. In contrast, IL-1β induces osteoblast differentiation from mesenchymal stem cells by activating the non-canonical Wnt (Wnt5a/ROR2) signaling pathway. However, during inflammation, IL-1β and TNF-α suppress osteoblastic differentiation by downregulating the expression of osteoblastogenic genes, such as runt-related transcription factor 2 (RUNX2), osterix (OSX), ALP, and COL1A1 in mesenchymal stem cells. The expression of these osteoblastogenic markers can be induced by IL-37, and IL-18 can enhance osteoblast proliferation [28].

In addition to a direct role in OS cell proliferation/differentiation, IL-1 members have been described to play a role in shaping the TME.

OS are characterized by a complex and plastic TME that includes, in addition to a mineralized extracellular matrix (ECM), various amounts of transformed osteoblasts, activated osteoclasts deriving from the monocyte–macrophage lineage, osteocytes, stromal cells, i.e., mesenchymal stem cells and fibroblasts, vascular cells as endothelial cells and pericytes, and immune cells of myeloid and lymphoid origin. The immune infiltrate and TME transcriptional profile can be used to stratify OS patients into prognostic groups, where patients having tumors characterized by the innate immunity signature have better survival compared to patients whose tumors show the angiogenic, osteoclastic, adipogenic, and upregulated PPARγ pathway signature [29].

With some heterogeneity across different OS tumors, malignant osteoblastic cells and myeloid cells are the most represented elements [30,31,32,33]. The variegated cellular microenvironment is reflected by a complex network of factors and receptors. Inside OS tumors, multiple ligand–receptor interactions occur, also involving IL-1β [30,31]. An analysis of OS tumor infiltrates by scRNA-seq revealed that the majority of TAMs in OS exhibited M2 polarization [32], and they may support immune escape and metastatic progression by secreting various immunosuppressive and angiogenic factors, such as IL-1β, IL-4, IL-10, and TGF-β. Han and collaborators [34] demonstrated that M2 macrophages sustain OS metastatic dissemination by secreting IL-1β, which activates the NF-κB/miR-181α-5p/RASSF1A/Wnt signaling pathway in malignant osteoblasts. In OS cells, the IL-1β/NF-κB signaling pathway is highly activated and positively correlated with the expression of miR-181α-5p, which can negatively regulate Ras-associated domain family protein 1 isoform A (RASSF1A). RASSF1A is a RAS-GPT binding protein that acts as a scaffold for many signaling pathways and behaves as a tumor suppressor gene inhibiting cell cycle progression and promoting apoptosis, reducing invasion and migration [35,36]. RASSF1A is frequently inactivated in most human cancers, it is poorly expressed in OS cell lines and in OS tumor samples, and its absence correlates with disease severity [37,38].

After the addition of an M2-TAM supernatant or exogenous IL-1β in OS cultures, the expression of NF-κB and miR-181α-5p increased. The expression of RASSF1A was suppressed and resulted in the accumulation of β-catenin, an index of activation of the Wnt/β-catenin pathway. The use of an IL-1R inhibitor (Anakinra) or NF-κB inhibitor (Rocaglamide A) and miR-181α- 5p knockdown were able to reverse Wnt/β-catenin activation and inhibit invasion and migration [34].

Similar activity of the IL-1β/NF-κB signaling pathway involving the NF-κB/miR-181α-5p/RASSF1A/Wnt-β-catenin pathway can also be hypothesized in other pediatric solid tumors, including EWS [39,40,41]. The role of IL-1β and Wnt signaling has also been implicated in the bone metastatic growth of breast cancer [21], which will be discussed in the section dedicated to the IL-1 family in bone metastasis. In OS cell lines, IL-1β, through the induction of NF-κB, also upregulates miR-181b expression, which, by repressing phosphatase and tensin homolog (PTEN) expression, promotes OS cell proliferation [42]. Additionally, IL-1β has been implicated in OS drug resistance. In a cohort of OS patients, the levels of IL-1β, IL-1R, and IL-1RAP expression in tumor samples were directly correlated with the presence of metastases and inversely correlated with the degree of tumor necrosis after neoadjuvant chemotherapy. Macrophages were able to reduce the sensitivity of OS cells to neoadjuvant chemotherapy drugs by secreting IL-1β, both in vitro and in vivo, and treatment with IL1Ra reversed the effect in vitro [43]. Moreover, in response to doxorubicin, IL-1β signaling can upregulate the expression of PD-L1 in OS cells, favoring immune evasion [44]. A comparison of ligand–receptor interactions between naive and treated patients showed that chemotherapy can induce a strong remodeling of the TME, with a decrease in immune cell communication and an increase in interactions between malignant osteoblasts [30].

In summary, these data indicate that IL-1β and M2 TAMs can support OS tumor growth, malignancy, immune escape, and drug resistance by activating multiple cell-intrinsic and -extrinsic mechanisms.

## 4. IL1 Family in Ewing Sarcoma

Recent results have highlighted that IL-1RAP activities are not limited to the roles played in the context of the IL-1 family in immune cells as IL-1R co-receptors. By transforming fibroblasts with multiple oncogenes, IL-1RAP has emerged as a protein that is directly induced by the fusion oncoproteins EWS::FLI1 (but also EWS::ERG) that drive EWS [45,46]. EWS::FLI1 induced IL1RAP expression by recruiting the BAF/p300 complex to the *IL1RAP* enhancer [45,46]. An analysis of published RNA-sequencing datasets and the detection of IL-1RAP protein expression in human tumor samples, xenografts, and cultured EWS cell lines demonstrated that IL-1RAP was expressed at high levels in EWS and that high IL-1RAP expression was significantly correlated with lower event-free survival in three different EWS cohorts [46]. IL-1RAP also emerged as a highly expressed molecule in EWS in another recent study looking for actionable immunotherapeutic targets [47].

In EWS cells, IL1RAP knockdown had no effect on cell growth in 2D conditions but strictly impaired anchorage-independent growth [46]. To the best of our knowledge, IL-1RAP knockdown in OS cells has not been studied yet, but it could be interesting to evaluate the presence of a similar inhibitory effect in 3D growth conditions. In a model of EWS xenograft growth and spontaneous metastasis based on TC32 EWS cells in immunodeficient mice, IL-1RAP knockdown significantly inhibited primary tumor growth, decreased invasion into surrounding normal tissues, and reduced metastatic dissemination to the lung and the metastatic burden [46]. Therefore, IL-1RAP was identified as a driver of metastasis in EWS. When looking for underlying mechanisms in EWS cells, IL-1RAP reveled unexpected functions, independent of IL-1R. Indeed, IL-1RAP was involved in the suppression of anoikis, a mechanism of programmed cell death that occurs when cells detach from the extracellular matrix and that cancer cells must overcome to enable the metastatic process. In EWS cells, IL-1RAP provides a cytoprotective mechanism by giving the cells the ability to maintain redox homeostasis and replenish the cysteine and glutathione (GSH) pools, even when extracellular uptake is restricted, during the metastatic cascade [45,46].

Cysteine is the substrate for GSH synthesis. The intracellular cysteine pool is derived from exogenous uptake through the xCT/CD98 transporter (xCT, also known as solute carrier family 7A11, SLC7A11, and CD98, also known as solute carrier family 3 member 2, SLC3A2) or de novo synthesis via the transsulfuration (TSS) pathway, which converts serine and methionine into cysteine and subsequently into the antioxidant glutathione. Of note, the xCT/CD98 transporter has also emerged as a marker of cancer stem cells in various tumor types, as it is highly expressed in tumor spheres [48], which suggests that cancers of different origins upregulate and utilize the same system for survival under conditions of anchorage-independent growth. IL1RAP contributes to the maintenance of the intracellular cysteine pool by binding the xCT/CD98 transporter system on the cell membrane to enhance exogenous cysteine uptake. When xCT functions are reduced due to cystine depletion, IL1RAP induces cystathionine gamma lyase (CTH) for de novo cysteine synthesis. Therefore, IL1RAP maintains the cysteine and glutathione pools, which are essential for redox homeostasis, protection from ferroptosis (iron-dependent cell death induced by cystein depletion), and anoikis resistance [45,46]. Interestingly, high CTH expression was found to be associated with shorter event-free survival in two separate cohorts of patients with Ewing sarcoma, supporting a role for CTH in Ewing sarcoma malignancy [46].

IL1RAP has previously been shown to be overexpressed in acute myeloid leukemias (AML) and other myeloid malignancies. In AML, IL1RAP enhances multiple oncogenic signaling pathways. In AML, IL1RAP physically interacts with and mediates signaling through FLT3 and c-KIT, two receptor tyrosine kinases that induce JAK2/STAT phosphorylation and AKT and MAPK activation in response to the FLT3 ligand (FLT3L) and stem cell factor (SCF), respectively, thereby increasing proliferation, invasion, and migration [49].

IL-1RAP is used in AML as an adaptor-molecule-dependent mechanism to amplify signaling that is not the result of a direct activating mutation in the receptors [49].

Based on these additional cell-intrinsic functions of IL-1RAP, the role of this protein in EWS could not be limited to anoikis suppression. Indeed, EWS shows high levels of FLT3 [50] and c-kit [51,52], which play a relevant role in migration and metastasis. EWS may exploit the same mechanisms of potentiation of oncogenic signaling demonstrated in AML as the activity of these receptors may be enhanced by interaction with IL-1RAP, which is also highly expressed in EWS. Although FLT3 and c-kit are not present in the interactome analysis of IL-1RAP in EWS [46], this hypothesis deserves further functional investigation. The cell-intrinsic roles of IL-1RAP in EWS are illustrated in Figure 1.

Additionally, IL-1RAP is involved in neural differentiation. Considering that EWS cells retain the potential for neural differentiation [53], the possible role of IL-1RAP in EWS neural differentiation should be investigated.

Since the expression of IL-1RAP was barely detectable in normal tissue in different organs, in both pediatric and adult subjects, the selective immunotherapeutic targeting of this protein in EWS may have the advantage of simultaneously affecting the signaling of different oncogenes, disrupting cysteine metabolism, reversing anoikis resistance, and inducing anti-tumor immune responses [45,46].

## 5. IL1 Family in Bone Metastasis

Bone metastasis occurs when cancer cells from primary tumors, such as breast, prostate, lung, and bone tumors, or other malignancies, spread to the bone. This process involves a complex interplay of various molecular and cellular factors, and emerging evidence suggests that IL-1 is actively involved in several mechanisms of this metastatic cascade [27,54,55,56].

IL-1 has been implicated in promoting the proliferation and invasiveness of cancer cells, facilitating the invasion of tumor cells into the bone microenvironment. Nutter and colleagues established, by serial selection, a clone of the human breast cancer cells MDA-MB-231 (MDA-IV) capable of specific homing and growth in the bone microenvironment after intravenous (i.v.) injection. Among the genes differentially expressed between MDA-IV and parental cells, the latter lacked the ability to home to the bone after i.v. injection (notably, parental cells can grow in the bone microenvironment after intracardiac injection, indicating differences specific to bone homing rather than colonization in the bone), the authors identified the higher expression of IL-1β in MDA-IV cell lines, associated with the cells’ ability to home to the bone microenvironment. Furthermore, by analyzing patient tumor samples, the researchers discovered that the increased expression of IL-1β in primary tumors was strongly correlated with adverse clinical outcomes and a significantly increased likelihood of developing bone metastasis [57]. In a later study, the authors showed that breast cancer cells engineered to overexpress IL-1β (MDA-MB-231- IL1B+) had enhanced bone-homing capabilities [56]. In both studies, the inhibition of IL1R signaling with the IL1R antagonists anakinra or canakinumab prevented or reduced metastasis [56,57].

In breast cancer, dormant cancer cells can give rise to bone metastases. In vitro, it was observed that long-term bone-marrow-conditioned medium was able to induce the increased formation of mammospheres. The inhibition of IL-1β with neutralizing antibodies prevented mammosphere formation in vitro and inhibited metastasis formation to bone in vivo. The mechanism was mediated by Wnt ligand secretion and autocrine Wnt signaling in tumor cells through the activation of NF-kB and CREB signaling. Treatment with vantictumab, which inhibits Wnt signaling by targeting the Frizzled receptors on cancer cells, was able to inhibit bone metastasis. In breast cancer, microenvironmental IL-1β of the bone metastatic niche or the endogenous production of IL-1β by breast cancer cells prompted breast cancer stem cells to form colonies through the activation of NF-kB and CREB signaling, Wnt ligand secretion, and autocrine Wnt signaling in tumor cells [21]. Moreover, contact between tumor cells and osteoblasts or bone marrow cells can increase IL-1β secretion from all three cell types [56].

Analogous results were obtained in prostate cancer models, in which the overexpression of IL-1β increased bone metastasis [54,58].

Human prostate cancer (PCa) cells secreting IL-1β can convert the bone stroma into a metastatic niche by upregulating genes such as S100A4 and COX-2, which are involved in the formation of cancer-associated fibroblasts (CAFs). Several studies have reported that CAFs can be impaired by anakinra, resulting in a reduction in the tumor burden and bone metastasis [54,59].

In the context of bone metastasis, the induction of angiogenic activity may enhance the blood supply to the metastatic lesion in the bone. The inhibition of IL-1β signaling in the bone environment by various approaches, such as the knockout of IL-1R1, pharmacological blockade with an IL-1 receptor antagonist, or reduction of circulating IL-1β levels with an anti-IL-1β antibody, resulted in a strong reduction in bone metastasis. Of note, both a reduced number of blood vessels and a reduced vessel length were observed in the bone upon disruption of IL-1β signaling [56].

The formation of new blood vessels, or angiogenesis, is a critical step in tumor growth and metastasis. IL-1 is known to stimulate angiogenesis by promoting the expression of vascular endothelial growth factor (VEGF) and other pro-angiogenic factors [54,60,61]. An additional mechanism is represented by the IL-1β-induced upregulation of pentraxin 3 (PTX3), an essential molecule in tissue repair, able to induce endothelial cells’ proliferation, migration, and tube formation [60,62].

Together with IL-1β, other IL-1 family members, such as IL-18 and IL-33, may play a role in inducing angiogenic and neovascular formation depending on the tissue context [60].

IL-1 plays an important role in the regulation of bone remodeling. It can stimulate the differentiation and activation of osteoclasts, the cells responsible for bone resorption. In the context of bone metastasis, cancer cells can take advantage of this property of IL-1 to induce excessive bone resorption, thereby creating a favorable microenvironment for their growth and survival within the bone. Osteoclast activation is achieved by the induction of IL-17, which acts first on osteoblasts, stimulating both COX-2-dependent PGE 2 synthesis and osteoclast differentiation factor (ODF) gene expression. The induction of osteoclast precursor differentiation into mature osteoclasts is critical for osteoclastic bone resorption [63]. Indeed, breast cancer bone metastases were significantly increased in a mouse model of autoimmune arthritis [64], highlighting the critical role of inflammation in bone metastasis [56,65,66].

## 6. Therapeutic Implications

The most well-known IL-1-targeting agents include the recombinant form of the receptor antagonist IL-1Ra (anakinra), the anti-IL-1β mAb (canakinumab), the anti-IL-1α mAb (bermekimab, MABp1), the fusion protein consisting of the ligand-binding regions of IL1R1 and IL-1RAP linked to the Fc region of human IgG1 (rilonacept), and Nadunolimab (CAN04) [67], a fully humanized ADCC-enhanced IgG1 antibody that targets IL1RAP and blocks IL-1α and IL-1β signaling [2,12]. All these drugs have been investigated, andhave received formal approval from the drug regulatory agencies for several immunological disorders, and show a good safety profile at clinical level [2,12].

### 6.1. IL-1-Targeting Drugs

A large number of animal studies and clinical trials have been conducted employing IL-1-targeting drugs for cancer therapy against hematologic malignancies, non-small-cell lung cancer, metastatic breast cancer, colon cancer, pancreatic cancer, renal cell carcinoma, and melanoma [2,68,69], but not yet in bone sarcomas, with no trials underway (clinicaltrials.gov, accessed on 16 December 2023).

Chemotherapy and radiotherapy used against carcinomas and sarcomas can affect and act through inflammasomes and IL-1β signaling. Cytotoxic drugs, including doxorubicin, daunorubicin, melphalan, gemcitabine, fluorouracil, cytarabine, methotrexate, paclitaxel, etoposide, vincristine, and cisplatin, can activate inflammasomes and IL-1β-related pathways in cancer cells but also in myeloid cells in the TME [70]. NLRP3 is the most commonly found chemotherapy-induced inflammasome, which may increase or reduce the anti-tumor effects of cytotoxic drugs depending on the balance between two effects: on one hand, immune cell activation and pyroptosis (an inflammatory type of programmed cell death involving the activity of caspase-1 and gasdermin) [71]; on the other hand, the IL-1β-induced M2 polarization of TAM and IL-1β-induced epithelial–mesenchymal transition, promoting invasion and metastasis. In some studies, the outcome of chemotherapy has indeed been improved by the combination of cytotoxic drugs and IL-1β/IL-1R inhibitors, such as anakinra, limiting the pro-tumor effects of IL-1β [17,68]. The clinical benefit of using recombinant IL-1Ra for the treatment of metastatic cancer holds promise and warrants further investigation.

### 6.2. IL-1 Family Signaling Molecules as Immune Checkpoints

Many members of the IL-1 family and their signaling pathways are now reasoned as new immune checkpoints acting across innate and adaptive immunity.

IL-1R signaling requires the dissociation, phosphorylation, and activation of IRAK family members, and this process is negatively regulated by IRAK3, a pseudokinase that lacks catalytic function but stabilizes the myddosome, leading to decreased pro-inflammatory cytokine production and suppressed inflammation. In a cohort of advanced urothelial cancer patients treated with immune checkpoint blockade (ICB) therapy, high IRAK3 expression in myeloid cells significantly correlated with enriched anti-inflammatory pathways and a worse clinical response to ICB. Tumors with low IRAK3 levels showed a significant enrichment of M1-like macrophages, T follicular helper cells, activated CD4+ memory T cells, and memory B cells, whereas monocytes and mast cells were more abundant in IRAK3-high tumor samples [72]. In a preclinical setting, when studying IRAK3 function by germline gene deletion in immunocompetent mice, homozygous IRAK3 knockout did not induce abnormalities or autoimmunity. In IRAK3-KO mice, IRAK3 deficiency delayed the growth of various murine cancer cell lines, and ICB treatment induced superior growth inhibition compared to wild-type mice. In vivo in the murine model and in vitro in human myeloid cells, IRAK3 acted as a myeloid-cell-specific immune checkpoint [72]. Further studies are needed to explore the prognostic value of IRAK3 in other cancers. To the best of our knowledge, its role in the bone sarcoma response to ICB has not yet been fully investigated, but, given the prevailing presence of macrophages and myeloid cells in bone sarcoma infiltrates, IRAK3 expression could be expected to have a role. To target the intracellular non-catalytic IRAK3 protein, selective chemical binders have been developed into proteolysis-targeting chimera (PROTAC) molecules [73]. The targeting of IRAK3 can also increase the production of the pro-inflammatory cytokine IL-12 [74]. The evaluation of IRAK3 PROTAC degraders in vivo in preclinical models may pave the way for a new generation of checkpoint inhibitors for different tumor types, especially for cold tumors showing a poor T and B cell infiltrate.

IL-1R8 has been proposed as a checkpoint regulating NK cell function in cancer [75,76]. IL-1R8 acts as a negative regulator, repressing inflammatory pathways, and as a co-receptor for the anti-inflammatory cytokine IL-37. Human and mouse NK cells express high levels of IL-1R8, which is also found in monocytes, T and B cells, dendritic cells, and epithelial cells in several tissues.

IL-1R8 inhibits IL-1-induced NF-kB and JNK activation by blocking the heterodimerization between IL-1R and IL-1RAcP and diminishing the recruitment of MyD88. The silencing of IL-1R8 in human NK cells led to increased NK cell activation, cytotoxicity, and degranulation, in response to IL-15 or IL-18 stimulation. In mouse models of lung and liver metastasis, the inhibition of IL-1R8 expression increased the NK cell number and hampered metastatic growth. Targeting IL-1R8 as a checkpoint of NK activity in tumors could be a valuable strategy against metastasis and further investigation is ongoing [75].

Another immune checkpoint related to the IL-1 family has been identified in the secreted IL-18-binding protein (IL-18BP). This IL-18 decoy receptor is frequently upregulated in different human and murine tumors, is detectable in both the TME and blood, and binds IL-18 with >3 orders of magnitude higher affinity than IL-18Rα. An engineered decoy-resistant IL-18 (DR-18), which retained its signaling potential but was resistant to interaction with IL-18BP, exerted a potent anti-tumor effect in a preclinical mouse model by promoting the development of polyfunctional effector CD8+ T cells. DR-18 also enhanced NK cell activity and maturation. These results highlight the potential of the IL-18 pathway for immunotherapy and implicate IL-18BP as a major therapeutic obstacle [25]. A functional human (h) DR-18 has been developed and it was able to elicit IFN-γ production in both human and cynomolgus macaque peripheral blood mononuclear cells, showing potential for clinical translation [25]. The relevance of IL-18BP in bone sarcomas has not yet been fully investigated but it could have a role in immune escape and ICB resistance.

### 6.3. Potential Targets of Engineered Immune Effectors

IL-1 family members include antigens that could be of interest for the generation of receptor-modified immune effectors for immunotherapeutic purposes. Human anti-IL1RAP antibodies are able to induce potent antibody-dependent cellular cytotoxicity against Ewing sarcoma cells [46]; for this reason, IL1RAP expression has gained attention from an immunotherapeutic perspective. Supporting elements are (i) the strong cell surface expression in selected cancers, such as acute [77] and chronic [78] myeloid leukemia and Ewing sarcomas [46]; (ii) the borderline presence in normal pediatric and adult tissues, including hematopoietic stem cells; (iii) the dependency of tumor cells for anoikis resistance, a compelling step in the metastatic cascade [46]. Chimeric antigen receptor T cells directed against IL-1RAP (IL-1RAP CAR T) were able to kill leukemic cells in vitro and in xenografts in preclinical mouse models of CML [78] and AML [77]. IL-1RAP CAR T have been proposed for a first phase I clinical trial in CML patients (NCT02842320, Study Details | Targeting Leukemic Stem Cell Expressing the IL-1RAP Protein in Chronic Myelogenous Leukemia (CML) | ClinicalTrials.gov
//clinicaltrials.gov/study/NCT02842320 accessed on 19 December 2023) and AML patients (NCT04169022, Search for: Other terms: NCT04169022 | Card Results | ClinicalTrials.gov /www.clinicaltrials.gov/search?term=NCT04169022, accessed on 19 December 2023) [12,79,80]. Since IL-1RAP is highly expressed also in EWS and may represent a dependency for metastatic progression, new preclinical studies directed towards evaluating the activity of IL-1RAP CAR T or NK against EWS have been initiated. Luo et al. [81] demonstrated the strong in vitro cytotoxic activity of IL-1RAP CAR NK cells against the EWS cell lines A673 and SKNMC, supporting further preclinical investigations in xenograft models of EWS.

The different targets under evaluation as immune checkpoints or for CAR T/NK cell development are summarized in Figure 2.

## 7. Conclusions

The IL-1 family includes several molecules that play a key role in cancer, in many immunological and cell intrinsic contexts, increasing proliferation, invasion, and anoikis resistance and enabling the metastatic cascade. Many of these molecules are therefore of high interest as actionable targets in neoplastic diseases, both hematological malignancies and solid tumors. In particular, three strategies are emerging as potentially relevant for tumors with still unmet clinical needs, such as bone sarcomas and bone metastasis therapies, either as single agents or in combination with conventional chemotherapies:Direct anti-metastatic drugs, represented mainly by IL-1β-targeting drugs that are aimed at reducing the cell-intrinsic activities of IL-1β and of M2 macrophage-produced IL-1β, boosting the pro-metastatic microenvironment in the TME and favoring proliferation, invasion, angiogenesis, and osteoclastogenesis.Immune checkpoints such as intracellular IRAK3 pseudokinase, actionable through PROTAC technology; IL-1R8, which has been identified as an immune checkpoint for NK activity; IL18BP, which can be overcome by the decoy-resistant molecule DR-18, which is able to avoid IL18BP binding and maintain the positive regulation of NK functions.Differentially and highly expressed antigens, such as IL-1RAP, for receptor-engineered immune effector-immunotherapy using IL-1RAP CAR T/NK cells. These strategies appear very promising and are entering preclinical and clinical development for CML, AML, and EWS.

## Figures and Tables

**Figure 1 cells-13-00233-f001:**
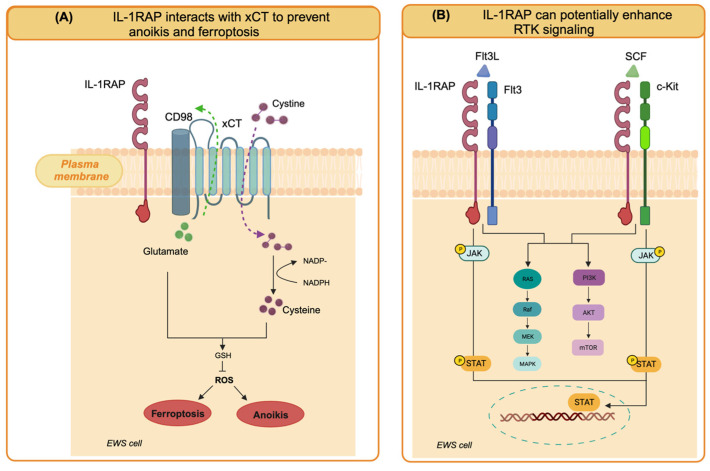
Cell-intrinsic functions of IL-1RAP in EWS cells. (**A**) IL-1RAP interacts with the xCT/CD98 transporter system on the cell membrane to enhance exogenous cysteine uptake and maintain cysteine and glutathione pools. By preventing the generation of reactive oxygen species (ROS), they confer resistance to anoikis and protection against ferroptosis. (**B**) IL-1RAP can potentially enhance oncogenic signaling pathways by interacting with RTK that are highly expressed in EWS cells, such as FLT3 and c-Kit (created with Biorender.com).

**Figure 2 cells-13-00233-f002:**
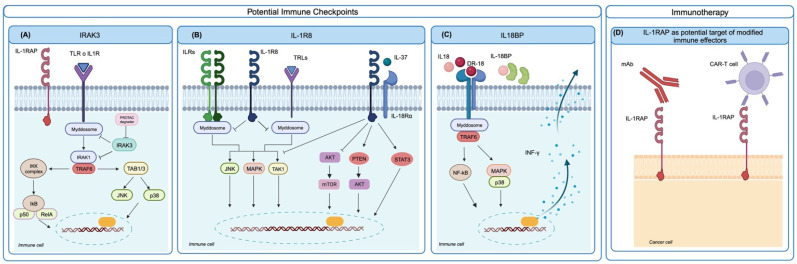
Potential IL-1-family-related immune checkpoints and targets for CAR T/NK cell development. (**A**) Upon IL-1R or TLR engagement, the IRAK3 pseudokinase restrains myddosome and IRAK1 activation and blunts inflammation. IRAK3 can be removed using PROTAC degraders. (**B**) IL-1R8 behaves as a negative regulator of IL-1R and as a co-receptor for the anti-inflammatory cytokine IL-37, acting as a major checkpoint for NK function. The targeting of IL-1R8 could increase NK activity. (**C**) IL18BP is a decoy receptor for IL-18 and can be bypassed by a decoy-resistant IL-18 (DR-18), which is able to avoid IL18BP and bind the proper activating receptor, increasing type 1 immune responses. (**D**) Overexpression of IL-1RAP on CML/AML or EWS cells can be exploited for immunotherapy using anti-IL-1RAP antibodies or engineered IL-1RAP CAR T/NK cells (created with Biorender.com).

**Table 1 cells-13-00233-t001:** IL-1 family members.

IL-1 Subfamilies	Cytokines	Pro-Inflammatory Receptor Complexes	Main Pro-Inflammatory Activities or Antagonist Activity	Additional Anti-Inflammatory or Decoy Receptor Complexes	References
IL-1 subfamily	IL-1α	IL-1R1-IL-1RAP	Role in innate and adaptive immunity, acute-phase response, antimicrobial defense mechanisms, integrity of epithelial barriers, and mucosal immunity.	sIL-1R	[1]
	IL-1β	IL-1R1-IL-1RAP	Differentiation and activation of myelomonocytic cells, neutrophils, monocytes, macrophages. In TME, IL-1β can have a role in tumor initiation, progression, angiogenesis, invasion, and metastasis. Induction of IL-17-mediated pathways and MDSC recruitment.	IL-1R2-IL-1RAP, sIL-R2	[1,6]
	IL-1Ra	IL-1R1-IL-1RAP	Antagonist function on the same IL-1R1 receptor.		[1,6]
	IL-33	IL-1R4-IL-1RAP	Induction of Type 2 immune responses. In TME, recruitment of Tregs and activation of M2 macrophages		[1]
IL-18 subfamily	IL-18	IL-1R5-IL-1R7	Type 1 immune responses. Stimulates IFN-γ production by Th1 and NK cells. Induces cytotoxic T-cell activation, promotes IgG2 production by B cells.	IL18BP, IL-1R8	[1]
	IL-37		Anti-inflammatory, suppression of innate immunity.	IL-1R5- IL-1R8	[1,7,8]
IL-36 subfamily	IL-36α, IL-36β, and IL-36γ	IL-1R6-IL-1RAP	IL-36 targets DCs, macrophages, NK, and T cells to enhance their function and anti-tumor immune responses. IL-36 also acts on stromal cells, which then produce chemoattractants to recruit immune cells.		[6,9]
	IL-36Ra	IL-1R6-IL-1RAP	Antagonist function on the same IL-1R6 receptor.		[9]
	IL-38	IL-1R1-IL-1RAP, IL-1R6-IL-1RAP	Antagonist functions with inhibitory effects on the Th17 and Th22 response.		[1,10]
	Unknown ligand	IL-1R9-IL-1R10	Proposed as alternative receptors for IL-18 in the brain or as synaptic adhesion proteins in trans-synaptic signaling.		[1,11]

## Data Availability

No new data were created or analyzed in this study.

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
