# Peer review of "IL-1 Family Members in Bone Sarcomas"

_cells, 2024, doi:10.3390/cells13030233_

Round 1
Reviewer 1 Report
Comments and Suggestions for Authors
In the review manuscript by Landuzzi et al., the authors describe the role of IL-1 family members in bone sarcomas and metastatic bone conditions with a focus on osteosarcoma and Ewing sarcoma.
The review deals with an interesting topic and gives a good overview of it.
Only a few discretional comments for minor edits are suggested for the authors.
In lines 22 and 42 "Ewing's sarcoma"was used, please replace it with "Ewing sarcoma" as it was in other places in the text and check if more of this form occurs in the text.
In Table 1 some formatting I needed to get a better alignment for the 5th column with the mentioned cytokines from the 2nd column.
IL-17 is not listed here and some others are discussed in the text, was there any special reason for that?
In line 170 reference 14 is discussed as one for degenerative diseases, however, the authors described it for chronic inflammatory diseases, of course, some overlap may exist between these entities but it would be good to add or replace it with the originally used chronic inflammatory disease term.
In line 526 for IL-1R8 the official name of IL1RAPL1 should be mentioned as well.
Author Response
Reviewer 1
Comments and Suggestions for Authors
In the review manuscript by Landuzzi et al., the authors describe the role of IL-1 family members in bone sarcomas and metastatic bone conditions with a focus on osteosarcoma and Ewing sarcoma.
The review deals with an interesting topic and gives a good overview of it.
We are grateful to the reviewer for the positive evaluation of our work.
Only a few discretional comments for minor edits are suggested for the authors.
- In lines 22 and 42 "Ewing's sarcoma” was used, please replace it with "Ewing sarcoma" as it was in other places in the text and check if more of this form occurs in the text.
The sentence was corrected throughout the paper (page 1, lines 22 and 42).
- In Table 1 some formatting I needed to get a better alignment for the 5th column with the mentioned cytokines from the 2nd column.
Table has been changed accordingly and a reference, that was forgotten, has been inserted (page 3: lines 101-102).
- IL-17 is not listed here and some others are discussed in the text, was there any special reason for that?
We did not list IL-17 or other cytokines outside the IL-1 family members in the table since they are not strictly members of the IL-1 family, but they are involved only in the activity of IL-1 family cytokines.
- In line 170 reference 14 is discussed as one for degenerative diseases, however, the authors described it for chronic inflammatory diseases, of course, some overlap may exist between these entities but it would be good to add or replace it with the originally used chronic inflammatory disease term.
The sentence was corrected accordingly (page 5, line 175).
- In line 526 for IL-1R8 the official name of IL1RAPL1 should be mentioned as well.
We have found that nomenclature about IL-1R8 is a little bit confusing because it was used as previous name of IL1RAPL1 now named IL-1R9. We added a sentence explaining the previous and current nomenclature in the paragraph dedicated to the description of IL-1 family receptor complexes (page 4, lines 147-148 and lines 151-155).
Reviewer 2 Report
Comments and Suggestions for Authors
Dear Editor,
Thank you for giving me an opportunity to review this manuscript about Il-1 family in sarcoma.
I suppose it is very informative and interesting for future immunotherapy application for sarcoma.
IL-1RAP inhibition seems promising target for sarcoma treatment because it involves multiple oncogenic pathways such as MAPK, NF-kB, Wnt pathways.
This manuscript summarizes functions of IL-1 family in sarcoma.
If possible, I would like the authors to discuss IL-1RAP knockdown effect in osteosarcoma and Ewing sarcoma.
Author Response
Reviewer 2
Comments and Suggestions for Authors
Dear Editor,
Thank you for giving me an opportunity to review this manuscript about Il-1 family in sarcoma.
I suppose it is very informative and interesting for future immunotherapy application for sarcoma.
IL-1RAP inhibition seems promising target for sarcoma treatment because it involves multiple oncogenic pathways such as MAPK, NF-kB, Wnt pathways.
This manuscript summarizes functions of IL-1 family in sarcoma.
We thank the reviewer for the appreciation of our work.
- If possible, I would like the authors to discuss IL-1RAP knockdown effect in osteosarcoma and Ewing sarcoma.
We have inserted a sentence illustrating the effects of IL-1RAP knockdown in EWS cells in vitro and in vivo (pages 8-9, lines 331-340). To the best of our knowledge IL-1RAP knockdown in OS cells has not been studied yet.
Reviewer 3 Report
Comments and Suggestions for Authors
This review summarizes the effects of IL-1 on osteosarcoma, Ewing's sarcoma and metastatic bone tumor. The paper covers the mechanism of IL-1 in these diseases, particularly exploring its potential as a targeted drug, immune checkpoint therapy, and more cutting-edge engineered immune therapy. This article gives us valuable insights into IL-1 of bone tumor, and has significant and distinct scientific value. It is well written, of high quality, and can be published in my opinion.
Comments on the Quality of English LanguageThere are still minor language writing flaws and potential optimization in the text. e.g. line 48-49, 384. More language flaws could exist.
Author Response
Reviewer 3
Comments and Suggestions for Authors
This review summarizes the effects of IL-1 on osteosarcoma, Ewing's sarcoma and metastatic bone tumor. The paper covers the mechanism of IL-1 in these diseases, particularly exploring its potential as a targeted drug, immune checkpoint therapy, and more cutting-edge engineered immune therapy. This article gives us valuable insights into IL-1 of bone tumor and has significant and distinct scientific value. It is well written, of high quality, and can be published in my opinion.
We thank the reviewer for the positive evaluation of our work.
Comments on the Quality of English Language
- There are still minor language writing flaws and potential optimization in the text. e.g. line 48-49, 384. More language flaws could exist.
The indicated sentences and some other mistakes have been corrected throughout the manuscript (page: 2, lines 46-49; page 11, lines 398-399 and 421).
The review article has been revised according to reviewers’ comments and suggestions.
We modified also “Funding” that was omitted (page 16, lines 630-631) and “Acknowledgement” where we inserted a sentence (see page 16, lines 633-634).